# Endophytic Fungi for Crops Adaptation to Abiotic Stresses

**DOI:** 10.3390/microorganisms12071357

**Published:** 2024-07-02

**Authors:** Adan Topiltzin Morales-Vargas, Varinia López-Ramírez, Cesar Álvarez-Mejía, Juan Vázquez-Martínez

**Affiliations:** 1Programa de Ingeniería en Biotecnología, Campus Celaya-Salvatierra, Universidad de Guanajuato, Mutualismo #303, Col. La Suiza, Celaya 36060, Mexico; 2Departamento de Ingeniería Bioquímica, TecNM/ITS Irapuato, Silao-Irapuato km 12.5, El Copal, Irapuato 36821, Mexico; 3Coordinación de Ingeniería Ambiental, TecNM/ITS Abasolo, Cuitzeo de los Naranjos #401, Col. Cuitzeo de los Naranjos, Abasolo 36976, Mexico; 4Departamento de Ingeniería Química, TecNM/ITS Irapuato, Silao-Irapuato km 12.5, El Copal, Irapuato 36821, Mexico

**Keywords:** endophytic fungi, abiotic stress, crop adaptation, symbiosis, agricultural sustainability, climate change

## Abstract

Endophytic fungi (EFs) have emerged as promising modulators of plant growth and stress tolerance in agricultural ecosystems. This review synthesizes the current knowledge on the role of EFs in enhancing the adaptation of crops to abiotic stress. Abiotic stresses, such as drought, salinity, and extreme temperatures, pose significant challenges to crop productivity worldwide. EFs have shown remarkable potential in alleviating the adverse effects of these stresses. Through various mechanisms, including the synthesis of osmolytes, the production of stress-related enzymes, and the induction of plant defense mechanisms, EFs enhance plant resilience to abiotic stressors. Moreover, EFs promote nutrient uptake and modulate the hormonal balance in plants, further enhancing the stress tolerance of the plants. Recent advancements in molecular techniques have facilitated the identification and characterization of stress-tolerant EF strains, paving the way for their utilization in agricultural practices. Furthermore, the symbiotic relationship between EFs and plants offers ecological benefits, such as improved soil health and a reduced dependence on chemical inputs. However, challenges remain in understanding the complex interactions between EFs and host plants, as well as in scaling up their application in diverse agricultural systems. Future research should focus on elucidating the mechanisms underlying endophytic-fungal-mediated stress tolerance and developing sustainable strategies for harnessing their potential in crop production.

## 1. The Plant Abiotic Stress Crisis

Abiotic stresses in plants refer to environmental factors that can negatively affect plant growth, development, and productivity. These stresses include extremes in temperature (both high and low), drought, flooding, salinity, heavy metals, pollutants, and nutrient imbalances [1,2]. Abiotic stress can disrupt normal plant physiological processes, leading to reduced crop yields, poor-quality crops, and sometimes even plant death. The impact of abiotic stress on plants has become a significant concern globally, particularly with the increasing frequency and intensity of extreme weather events due to climate change. These stresses can affect plants at various growth stages, from germination to maturity, and can occur individually or in combination, further exacerbating their effects [1].

Climate change and human activity significantly contribute to abiotic stresses such as alterations in temperature, precipitation patterns, and the frequency of extreme weather events, which affect crop production in several ways. Climate change leads to an increased frequency and intensity of extreme temperatures, causing periodic heat waves and cold snaps [3]. Both high and low temperatures can affect crop development; high temperatures can accelerate crop development and even kill the plants, while frost can severely damage crops, leading to a reduced yield and quality. In addition, changes in precipitation patterns result in water shortages and droughts, leading to wilting, stunted growth, and a decreased yield. Conversely, heavy rainfall and flooding can waterlog soils, resulting in oxygen deficiency in roots, nutrient leaching, and an increased susceptibility to pathogens. Climate change also increases the frequency and intensity of extreme weather events such as hurricanes and tornadoes; these phenomena can affect farmland by disrupt planting and harvesting schedules and bringing about post-harvest losses [3,4]. Another negative effect of climate change for crop production is the alteration to the distribution of invasive species and pests, since warmer temperatures and changing rainfall patterns create favorable conditions for pests and diseases to thrive, causing crop damage and yield losses [3,5]. Thus, the impact of abiotic stress on plants has become a significant concern globally, particularly with the frequency and intensity of extreme weather events due to climate change and human activity. The current and potential future impacts of plant abiotic stress are significant, affecting both agricultural productivity and food security worldwide [2].

Addressing the plant abiotic stress crisis requires urgent action and coordinated efforts at local, national, and international levels. The strategies to mitigate the impacts of abiotic stress on plants include developing stress-tolerant crop varieties, implementing sustainable and ecofriendly agricultural practices, improving water management, conserving the soil and biodiversity, and enhancing the resilience to climate change. In this review, we discuss the potential use of endophytic fungi (EFs) to ameliorate the impact of abiotic stress in crops facing climate change.

## 2. Endophytic Fungi in Perspective

EFs are a group of microscopic fungi that live inside the tissues of plants without causing any apparent harm to their host. They have gained significant attention in recent years due to their potential ecological, agricultural, and pharmaceutical applications [1,6]. EFs are a diverse group and can be found in various plant species worldwide. They have been found in almost every plant genus, from grasses to trees and from tropical rainforests to arid deserts [7].

The study of EFs has shed light on the complex interactions among plants, fungi, and the environment. EFs play a significant role in nutrient cycling, carbon sequestration, and the overall health of ecosystems. In many cases, EFs form mutualistic relationships with their host plants. They can provide benefits to their host, such as enhanced growth and an improved tolerance to environmental stress factors such as drought or salinity. In addition, EFs can help protect plants from herbivores and pathogens by increasing the plant resistance or by producing toxins that deter herbivores, while others compete with or inhibit plant pathogens [8,9].

Besides the effects of EFs on the development of plants, it is known that they can have significant effects on the microbial community of soils. EFs can enhance the release of root exudates, such as sugars, amino acids, and organic acids, into the soil [10]. These exudates serve as an energy source for soil microorganisms, stimulating microbial growth and diversity. Further, some EFs can suppress the growth of pathogenic microbes in the soil surrounding the roots by the release of antimicrobial metabolites, thus acting as biocontrol agents against these pathogens and modulating the composition and abundance of soil microbial communities [10,11]. Also, EFs and other endophytes can modify the soil structure by the production of extracellular polymeric substances, which contribute to soil aggregation and stability, thus shaping the overall microbial community structure of the rhizosphere [12].

Recent research has focused on EF bioprospecting for novel compounds and bioactive substances. These microorganisms represent a valuable resource for discovering new drugs, agricultural solutions, and biotechnological products. EFs are known to produce a wide range of bioactive specialized metabolites. These compounds have attracted attention for their potential use in pharmaceuticals, agriculture, and industry [13]. While the potential of EFs is exciting, there are challenges in harnessing their benefits. Research is ongoing into understanding their biology and ecology, the specific mechanisms of their interactions with plants, and how to effectively utilize them in agriculture, biotechnology, and medicine.

EFs can effectively improve the development and fitness of plants, including crops of economical and nutritional importance, by distinct mechanisms such as plant growth promotion, disease suppression, improved nutrient acquisition, allelopathy, herbivore and pathogen defense, and enhanced stress tolerance. In this review, we focused on discussing the role of EFs in ameliorating the impact of abiotic stress factors on crops.

## 3. Mechanisms of Endophytic Fungal Colonization of Plant Tissues

Given that the definitive characteristic of EFs is the colonization of plant tissues, it is imperative to delve deeper into this aspect. EFs colonize plant tissues through many complex mechanisms. Each colonization mechanism can vary depending on the specific fungal strain and the plant host, but there are some general processes involved in the colonization of plant tissues. At the beginning of a plant–fungal interaction, the EF typically enters the plant host through natural openings such as stomata, wounds, or root tips. Further, some fungi may produce specialized structures or adhesive molecules to attach to the plant surface [14].

How EFs locate and approach their host plant seems to be mediated by chemotaxis; however, further research is required. Some EFs can actively move toward chemical cues released by their host plant, guiding them toward potential entry points [15]. Once attached to the plant’s surface, EFs employ various strategies to penetrate the plant tissue. These can involve the secretion of enzymes, such as cellulases or pectinases, to break down the plant cell wall and gain entry [6,16]. Then, EFs can grow both intercellularly (between plant cells) and intracellularly (inside plant cells) [16]. Fungi that grow intracellularly may form specialized structures called hyphal coils or microsclerotia [17].

Successful endophytes often have mechanisms to evade or suppress the plant immune response. They can secrete molecules that inhibit plant defense mechanisms, making them less likely to be recognized as pathogens. Some EFs produce bioactive specialized metabolites, which can further contribute to their successful colonization of plant tissues [8,18].

EFs can persist within a plant for extended periods, potentially throughout the plant’s life cycle. They may spread throughout the plant and colonize various tissues, including the seeds, leaves, stems, and roots [19]. Therefore, EFs can be transmitted vertically (from parent to offspring) and horizontally (between plants) through various means, including through seeds, root-to-root contact, and aerial spore dispersal [20,21].

## 4. Endophytic Fungus–Host Plant Interactions

EF–host plant interactions can be as specific as a host species and an endophyte strain. Among these interactions, mutualistic relationships are interesting due to the various benefits to the plants. As part of these benefits, EFs can help plants acquire essential nutrients, such as phosphorus and nitrogen. EFs can access nutrients from the soil that may be otherwise unavailable to the plant and transfer these nutrients to their host [22]. In addition, some EFs produce specialized metabolites that have antifungal or antibacterial properties; this can help to protect the host plant from pathogen attacks. For example, the EFs of grasses can produce alkaloids that deter herbivores [23]. Further, EFs can contribute to the drought tolerance of host plants. EFs help plants by improving their water and nutrient uptake, which is especially valuable in arid or water-stressed environments [24]. EFs can interact with plants through chemical signaling and molecular mechanisms by releasing substances that modulate plant growth and development, contributing to their mutualistic effects [16].

Not all EF–plant interactions are mutually beneficial. Some endophytes can become pathogenic under certain conditions, causing harm to their host plants [9]. This EF beneficial/harmful duality makes the study of these interactions even more complex. Beyond the relationship between EFs and their host plants, this phenomenon has ecological implications, since EFs play a role in plant community dynamics and ecosystem diversity. Endophytic fungi can influence the composition of plant communities and the overall structure of ecosystems.

## 5. Endophytic Fungi and Host Plant Fitness

The relationship between EFs and host plant fitness is a subject of great interest in ecology and plant biology. The fitness of a host plant refers to its ability to survive, reproduce, and pass on its genetic material to the next generation [25,26]. EFs can have both positive and negative impacts on host plant fitness, depending on various factors [27].

On the beneficial side, EFs can enhance the uptake of essential nutrients, such as nitrogen and phosphorus, from the soil. This increased nutrient availability can lead to improved plant growth, which, in turn, can enhance the fitness of the plant by increasing its reproductive potential [8]. Also, endophytic fungi can help host plants tolerate environmental stress factors such as drought, a high salinity, or extreme temperatures. This stress tolerance can enable the plant to survive and reproduce under adverse conditions, thereby enhancing its fitness [28]. Furthermore, some EFs produce specialized metabolites that deter herbivores and protect the plant from pathogens. So, the reduction in herbivory and pathogen damage can lead to higher plant fitness by reducing the loss of resources and high-value tissues such as reproductive structures [29]. In addition, EFs can help plants compete with neighboring vegetation for resources. This competitive advantage can also lead to an increased fitness by allowing the plant to access more resources and grow more vigorously [19].

In terms of negative impacts on plant fitness, some EFs can turn pathogenic under certain conditions, causing harm to their host plant. Pathogenic endophytes can lead to decreased plant survival by causing diseases, reducing growth, or even killing the plant [30]. On the other hand, the resources invested by the host in maintaining endophytic associations might be diverted from other critical plant functions, which can affect the plant’s overall fitness. This is particularly relevant when endophytes do not provide substantial benefits [31]. While the advantages of a mutualistic plant–EF symbiosis have been extensively documented, the factors and mechanisms related to non-beneficial relationships, the balance of endophytic–pathogenic behavior, and the associated cost–benefit of an EF infection need more research. By understanding the molecular mechanisms, the genetic cues, and the physiological traits related to the process by which some EFs become pathogenic and vice versa, new insights for biotechnology and agroecology can be discovered.

Among the most studied and documented beneficial EFs are *Trichoderma* spp., particularly in mitigating abiotic stress and promoting plant growth. *Trichoderma* species are commonly used in agriculture as biocontrol agents against pathogens, but they also enhance plant growth by promoting root development and nutrient uptake [32]. *Trichoderma* spp. have been shown to improve plant tolerance to various abiotic stresses, including drought, salinity, and heavy metal toxicity. They achieve this through mechanisms such as the production of stress-related enzymes and phytohormones [32,33]. Besides *Trichoderma*, the other genera of EFs that are used in agriculture are *Clonostachys rosea* [34], *Piriformospora indica* [35], and some *Fusarium* species; however, some *Fusarium* species are causative agents of plant diseases, so they must be considered with the necessary precautions [36]. More research is needed to discover and characterize new beneficial endophytic fungi that can be used to help plants cope with abiotic stress factors.

## 6. The Impact of Endophytic Fungi on Abiotic Stress Adaptations in Crops

Among all the benefits associated with EF–plant interactions (Figure 1), the potential to enhance crop adaptations to various abiotic stresses, including drought tolerance, salinity resistance, and adaptations to temperature and pH changes, is of particular interest due to its implications for sustainable agriculture, food sovereignty, the eradication of hunger and malnutrition, and the care of the environment [1,2,6]. Strategies involving the use of specific EFs or the manipulation of their interactions with host plants through inoculation or genetic engineering hold promise in developing stress-tolerant crop varieties. However, practical applications in agriculture would require a deeper understanding of the mechanisms involved, the optimal conditions for the EF–plant interactions, and the potential impacts on the environment and ecosystem.

Most of the unfavorable conditions that a plant must resist, and that are derived from abiotic stress factors, include oxidative stress. Oxidative stress is related to the production of dangerous oxygen radicals or other molecules capable of removing electrons from biomolecules [37]. This imbalance in the redox status of biomolecules provokes their quick degradation, dramatically affecting the pathway through which the biomolecules are related. Most organisms have protection mechanisms against oxidative stress, and most of them related to the synthesis of free radical scavengers such as glutathione, flavonoids, and other antioxidant molecules [38]. EFs can provide protection to plants in high-oxidative environments, and thus, shield them against abiotic stress.

Next, we discuss some of the benefits provided by EFs to the abiotic stress tolerance of crops.

### 6.1. Role of Endophytic Fungi in Salt Stress Amelioration

Among all the abiotic stresses, soil salinization is the most harmful. Soil salinization includes saline and alkaline soil exposure, which are both defined as a high salt concentration. Salinization is commonly linked to a high sodium cation (Na+) concentration and a high pH, often due to an elevated alkali concentration [39]. Otherwise, soil is classified as saline if its osmotic pressure is more than 0.2 MPa and its electrical conductivity is greater than 4 dS/m, corresponding to around 40 mM NaCl [40]. Soil salinization can occur due to both natural factors and human activities, which lead to the accumulation of soluble salts in the soil layers. The presence of dissolved ions in water can have a direct impact on the growth of crops; for example, the cations attached to soil particles can influence the soil structure, indirectly affecting crop productivity. According to a Food and Agriculture Organization (FAO) report from the year 2000, the global extent of salt-affected soils, which includes both saline and sodic soils, was estimated to cover an area of 831 million hectares. Experts estimate that, by 2050, soil salinity will have damaged at least 50% of the world’s farmland [41]. In addition to this, climate change is increasing the concerns related to soil salinization. As stated by predictions of primary soil salinization under climate change, the following regions are considered hotspots for soil salinization: South America, Southern and Western Australia, Mexico, Western United States, and South Africa [42]. Soil salinity has several negative impacts on crop productivity, leading to low economic returns and soil erosion. Salinity affects various aspects of plant development, including germination, vegetative growth, and reproductive development. It imposes ion toxicity, osmotic stress, nutrient deficiencies, and oxidative stress on plants, limiting water uptake from the soil. Salinity also disrupts the nutrient balance and uptake in plants. It adversely affects photosynthesis, reduces the leaf area and chlorophyll content, and inhibits reproductive development. Salinity can also hinder seedling growth; enzyme activity; and DNA, RNA, and protein synthesis. Overall, soil salinity has detrimental effects on plant growth and development, leading to reduced crop productivity [41,43].

The severity of the impact depends on the duration and intensity of salt stress, as well as the sensitivity of the plant species or variety. To mitigate the negative effects of salt stress on plant growth and yield, strategies such as the use of salt-tolerant crop varieties, the addition of soil amendments, and the inoculation of beneficial microorganisms such as fungal endophytes are being explored [6]. Fungal endophytes play a crucial role in enhancing the salt tolerance of plants. They can withstand high salt concentrations and are able to colonize the roots of plants. When inoculated on salt-sensitive plants, fungal endophytes have been shown to promote germination and improve biomass-related parameters under salt stress conditions. They achieve this by increasing the host plant biomass and nutrient acquisition rate, altering the root architecture, carrying out osmotic adjustments, and alleviating oxidative stress in the host cells. Overall, fungal endophytes contribute to the salt stress tolerance of plants by enhancing their growth and mitigating the negative effects of salt on plant development [6,44].

For example, the fungal endophytes *Ulocladium* sp., *Fusarium avenaceum*, *Chaetomium* sp., and *F. tricinctum* have been found to assist plants in coping with salt stress. These endophytes were isolated from plants in the Northwestern Himalayan region and have shown the ability to impart salinity stress tolerance to mung bean plants (*Vigna radiata* L. Wilczek). *F. avenaceum* was found to be superior in inducing salinity stress tolerance and increasing the yield of mung bean plants. These endophytes help their host plants survive under salt stress conditions by reducing the deleterious effects of salt stress through various mechanisms, such as scavenging reactive oxygen species (ROS); maintaining ion homeostasis; and accumulating organic solutes, sugars, proteins, and lipids [45]. Table 1 describes some EF–plant interactions with a positive impact on plant fitness.

### 6.2. Endophytic Fungi Increase Tolerance of Crops to Temperature Stress

As mentioned, the lack of plant adaptations to abiotic stress could result in crop losses and soil deterioration. The use of fungal endophytes to increase the yield production and plant stress tolerance could be an ecofriendly alternative for this purpose. EFs can mitigate plant abiotic stress by their ability to colonize plant tissues, secondary metabolite synthesis, and the induction of the plant’s systemic defense [6,7,46].

Heat stress is defined as an increase in the ambient temperature up to 10–15 °C. The level of damage in plant tissues caused by heat stress involves physiological and cellular changes, which affect the stability of proteins and cellular components [47]. Under heat stress, an intricate biochemical and molecular response is activated that involves a sequence of processes from signal perception up to the expression of heat shock proteins (HSPs), ROS, and antioxidant enzymes [48,49]. Some biochemical responses involve a disruption of photosystems, rubisco heat-degradation activation, and chlorophyll reduction [50]. A common response during drought and heat stress is the upregulation of the transcription factor APETALA 2 (AP2)/ethylene-responsive factor (ERF) [51]. Likewise, it has been observed that heat stress can generate transcriptional memory after its induction, at least in the ascorbate peroxidase 2 (*APX2*) gene and the histone H3K4 hypermethylation, to produce a chromatin modification [52]. Plants can naturally face heat stress by a network pulse model that provokes a signal cascade, enabling stress tolerance and acclimatization [53]. Further, some authors have suggested that plants can ameliorate heat stress by the overexpression of rubisco activase (Rca) to offset the heat-induced rubisco inhibition [54].

The use of fungal endophytes for heat stress tolerance has been evaluated in different crops; for example, a study in soybeans and sunflowers using *Aspergillus flavus* as an endophyte found that this symbiosis mitigated heat stress and significantly enhanced the plant’s synthesis of proline, abscisic acid, and phenolic compounds in comparison with controls at 40 °C [55]. Similar results were observed in the same plant seedlings inoculated with *Rhizopus oryzae* and exposed to temperatures of 25 °C and 40 °C, with both species showing high levels of ascorbic acid oxidase (AAO), catalase (CAT), proline, phenolics, flavonoids, sugars, proteins, and lipids. In this study, it was also detected that the EFs increased the chlorophyll content, shoot and root lengths, and biomass compared to control plants [56]. The accumulation of specialized metabolites and the enzymatic activity upregulation of the enzymes involved in oxidative stress metabolism is a feature observed when plants grow under heat stress. Endophytes could stimulate this beneficial process, as has been described in cucumbers inoculated with *Thermomyces* sp. and sunflowers and soybeans inoculated with *Asperguillus niger* [57,58]. Furthermore, in geothermal soil simulation experiments, tomato plants and panic grass were evaluated in the presence of *Curvularia protuberata*, which was found to improve the level of drought and heat tolerance of the plants [59].

**Table 1 microorganisms-12-01357-t001:** Positive interactions between fungal endophytes and their host plants in response to abiotic stress.

Symbiotic Relationship (Endophytic Fungus/Plant)	Benefit for Plant Host	Abiotic Stress	Reference
*Piriformospora indica*/barley	Enhanced growth and yield; improved efficiency of water use and nutrient uptake; increased antioxidant capacity.	Salt stress	[60]
*Aspergillus ochraceus*/barley	Enhanced antioxidant capacity; protection against fungal pathogens; production of IAAs.	Salt stress	[61]
*Stemphylium lycopersici*/maize	Improved chlorophyll a/b ratio; increase in production of carotenoids and specialized metabolites; enhanced antioxidant and enzymatic activities; lipid peroxidation reduction; mitigation of ionic ratio misbalance.	Salt stress	[62]
*Penicillium minioluteum*/soybean	Increased shoot length and biomass production; chlorophyll and flavonoid content increase; leaf area increase; improved nitrogen uptake; regulation of hormone production.	Salt stress	[63]
*Periconia macrospinosa*, *Neocamarosporium chichastianum*, *Neocamarosporium goegapense*/cucumber, tomato	Increased chlorophyll concentration, leaf proline content, and enzymatic activity.	Salt stress	[64]
*Thermomyces* sp./cucumber	Heat stress tolerance; enhanced photosynthesis, water use efficiency, root length, and induced antioxidant enzyme activities; increased metabolite pool.	Temperature stress	[57]
*Paecilomyces formosus*/japonica rice	Improved plant height, fresh weight, dry weight, and chlorophyll content; heat stress tolerance.	Temperature stress	[65]
*Rhizopus oryzae*/sunflower and soybean	High levels of ascorbic acid oxidase, catalase, proline, phenolics, flavonoids, sugars, proteins, and lipids; enhanced chlorophyll content, shoot and root lengths, and fresh and dry biomass; tolerance to thermal stress at 40 °C.	Temperature stress	[56]
*Piriformospora indica*/grapevine	Reduced leaf electrolyte leakage, lipid peroxidation, and ROS content; increase in leaf abscisic acid, polyamines, soluble carbohydrates, proline, soluble proteins, and total phenolics; tolerance to cold stress.	Temperature stress	[66]
*Penicillium rubens* and *P. bialowienzense*/highbush blueberry	Higher photochemical sufficiency and lower oxidative stress than uninoculated controls; tolerance to cold stress.	Temperature stress	[67]
*Aspergilus welwitschide*/soybean	Enhanced production of phytohormones and solubilized inorganic phosphates; higher root length and fresh/dry weight than control; strengthened antioxidant system.	Toxic metal oxidative stress	[68]
*Trichoderma* spp./maize, tomato, cucumber	Enhanced root development, leading to better water and nutrient uptake.	Drought stress	[69]

Temperature stress also includes cold stress. It is known that cold induces a series of morphological and physiological changes in plants, affecting the lipid composition, membrane fluidity, differential flow of Ca^2+^, protein turnover, and induction of oxidative stress [70,71]. Several studies have shown a positive influence of fungal endophytes on the cold stress tolerance of plants. For example, the symbiosis between Arabidopsis and the fungus *Piriformospora indica* increases the plant’s expression of C-repeat binding factors (CBFs) and cold-regulated genes (CORs), triggering cold acclimatation [72]. In *Hordeum vulgare*, it has been demonstrated that *P. indica* acts positively on the plant’s nutrient uptake under cold stress [73]. Studies in bananas (*Musa acuminate*, variety Tianbaojiao) at 4 °C with the roots colonized by *P. indica* have demonstrated a reduction in the malondialdehyde (MDA) and hydrogen peroxide (H_2_O_2_) concentrations, an increase in the superoxide dismutase (SOD) and catalase (CAT) activities, and the augmentation of soluble sugars (SS) and the proline content [74].

In addition, the use of endophytes is not limited to monocultures. In a polyculture study using lettuce, chard, and spinach and a combined inoculum of *Penicillium fuscoglaucum* and *P. glabrum* as the endophytes, an increase in the plants’ nutritional status and the augmentation of the antioxidant compound (flavonoid) content was observed compared with uninoculated plants under temperature stress [75].

### 6.3. Crop Protection against Toxic Metal Oxidative Stress by Endophytic Fungi

As mentioned before, the interaction between the host plant and endophytes provides many benefits for both. This interaction allows plants to colonize and survive in niches where other plants cannot [76]. Among all the sources of oxidative stress that affect plant development, the oxidative effect of soils polluted with toxic metals (TMs) is of special interest due to increasing TM pollution. Pollution with TMs has been augmented because of industrial activities and mining extractions, which result in a high accumulation of minerals and other chemicals in the environment [6,77]. Among the problems caused by TM pollution, the negative impact on crop production is an obstacle to the eradication of hunger and malnutrition. The toxicity and oxidative activity of TMs is dependent on their reactivity and oxidation state, which is related to the soil pH, organic matter, and presence of ions. One option for reducing the oxidative stress related to TMs is by using fungal endophytes. EFs colonize plant tissues without producing notable symptoms and protect the plant against the oxidative effects of TMs [78].

Fungal endophytes can protect plants against TMs in many ways. One of these mechanisms is mediation by metal flocculants or chelating agents. Endophytes produce exopolysaccharides, lipids, glycolipids, and proteins that can capture and flocculate metal ions. The increased plant tolerance to TMs in the presence of endophytic fungi is related to metal ion biosorption and immobilization, modifications to the metal redox status, and the production of biopolymers and siderophores, which chelate the toxic metal ions, preventing the associated oxidative stress [79]. In addition to these mechanisms, as mentioned before, endophyte–host plant interactions encourage a better nutrient uptake, promoting a better plant response to oxidative stress. An example of this is the endophyte *Sporobolomyces ruberrimus*; this basidiomycete protects its host using siderophores, and promotes the development of lateral roots. *S. ruberrimus* chelates excess Fe, but does not interfere with the uptake of other micronutrients [80].

Plants modulate their abiotic stress response using mechanisms such as phytohormone signaling. Cytokinin, abscisic acid (ABA), ethylene, inositol phosphate, and NO_2_ are some examples of the molecules that regulate the response against ROS produced during abiotic or biotic stress [38]. Endophytes can promote or regulate phytohormones in their host, as has been described for *Sporobolomyces rubberriums*, which can regulates the ethylene concentration to resisting the oxidative stress produced by Fe [80]. Some endophytes can produce phytohormones or regulate those from their host, increasing the levels of plant antioxidant enzymes or antioxidant molecules [81]. One example is the production of organic acids, such as ascorbic acid, as an antioxidative strategy to neutralize the heavy metal effect and decrease the ROS effect. Auxins (IAAs), cytokinin, ethylene, gibberellin (GA), and ABA are essential for adaptive responses in an oxidative stress environment. Related to this, endophytes produce metabolites that stimulate and regulate host hormone production, increasing the plant’s metabolic response against abiotic stress [81].

### 6.4. Endophytic Fungi and Crop Drought Tolerance

The symbiotic association between EFs and plants has garnered significant attention in recent years due to its potential application in sustainable agriculture, particularly in the face of climate-change-induced drought events. Understanding the mechanisms underlying the drought tolerance conferred by EFs is crucial for harnessing their full potential in agricultural systems. EFs can promote drought tolerance in host plants by enhancing water uptake and retention through various mechanisms [82]. Additionally, fungal hyphae can penetrate deep into the soil, accessing water reservoirs that are otherwise unavailable to plant roots, thus ensuring water availability during periods of drought stress [83]. Furthermore, EFs such as Epichloë endophytes can modulate the levels of plant hormones such as abscisic acid (ABA) and cytokinins, which play crucial roles in regulating plant responses to drought stress [84]. For instance, some EFs produce ABA analogs that mimic the effects of endogenous ABA, thereby inducing stomatal closure and reducing water loss through transpiration [84,85]. Conversely, other EFs produce cytokinins that promote cell division and expansion, thereby enhancing the resilience of plant tissues to drought-induced damage [86]. As with the other abiotic stresses, drought can lead to the accumulation of ROS in plant cells, causing oxidative damage to the cellular components. EFs can mitigate this oxidative stress by activating antioxidant defense systems in their host plants. For example, certain fungi produce enzymes such as superoxide dismutase and catalase, which scavenge ROS and protect plant cells from oxidative damage [86]. Additionally, EFs can enhance the accumulation of compatible solutes such as proline, which act as osmoprotectants and help maintain cellular homeostasis under drought conditions [24].

The ability of EFs to enhance the drought tolerance of crops holds great promise for sustainable agriculture, particularly in regions prone to water scarcity. By inoculating crop plants with drought-tolerant EFs, farmers can potentially reduce their reliance on irrigation and chemical inputs, thereby promoting environmental sustainability and resilience to climate change. Furthermore, the use of EFs as biofertilizers and biocontrol agents can enhance the productivity and resilience of agricultural systems. Despite the significant potential of EFs in enhancing the drought tolerance of crops, several challenges remain to be addressed. These include the identification and characterization of drought-tolerant EFs suitable for different crop species and environmental conditions, as well as the development of efficient inoculation methods for large-scale agricultural applications [87]. Additionally, further research is needed to elucidate the interactions between EFs and other beneficial microorganisms in the rhizosphere, as well as their long-term effects on soil health and ecosystem functioning.

## 7. Perspectives for Future Research and Development of Methods for Exploiting Beneficial Endophytic Fungi

The benefits of using endophytes for enhancing the abiotic stress tolerance of crops to address climate change are clear; however, there is still a long way to go, especially with respect to the discovery of new EF strains that can be used for different crops and under changing environmental conditions. Thus, future research on this topic should focus on the study of the interaction mechanisms of EFs and crops in abiotic stress environments by combining various methods such as microbiology, genetics, molecular biology, metabolomics, and ecology. The goal should be to develop protocols for the rapid isolation and identification of stress-tolerant/-resistant EF strains and methods for the characterization of the molecular mechanisms by which EFs exert their beneficial action. Also, it is important to characterize the genetic and enzymatic pathways that EFs use to survive in unfavorable environmental conditions and to interact with crops to establish beneficial symbioses.

To achieve this objective, microbiological methods must be applied to isolate, cultivate, and store EF stains. Since this has the potential to counteract some of the effects of climate change on food production, food safety, and hunger eradication, it is mandatory to generate EF collections with free global access, especially for vulnerable populations [88].

For a better understanding of the cellular interactions between EFs and their host plants, cutting-edge microscopic methods should be used, for example, Raman/IR microscopy and photon microscopy [16,89,90]. Further, the use of molecular techniques such as DNA sequencing, metagenomics, transcriptomics, and proteomics is necessary for a fine and detailed characterization of the evolution, phylogeny, genetic and functional potential, and gene expression and protein profiles of beneficial EFs [91,92]. Also, these techniques can be applied to the study of the genetic, functional, physiological, and systemic responses of plants in a symbiosis with EFs, paying attention to the improvement in plant fitness and growth, crop productivity, stress tolerance/resistance, and resilience in different environmental settings.

In addition, standard protocols for functional assays such as in vitro analyses, greenhouse experiments, and field studies must be developed to study specific EF–plant interactions, such as the production of plant-growth-promoting compounds, pathogen biocontrol activity, nutrient exchange, and resistance to biotic and abiotic stresses. These protocols would be developed with the aim of establishing a world EF collection of strains with proven characteristics to improve plant growth and crop productivity, and would even include information about the crop, environmental conditions, and effects for which each EF strain could be used.

Other methods that could be used to improve the interaction between EFs and crops under abiotic stress are bioinformatics and genetic engineering. Using bioinformatic tools, it is possible to analyze omics data to identify the genes, metabolic pathways, and regulatory networks involved in the beneficial interactions between EFs and crops, but also the molecular basis by which EFs tolerate/resist unfavorable environmental conditions [93]. Further, bioinformatics could be useful for predicting the potential benefits and the ecological roles of specific EF strains based on genomic, transcriptomic, proteomic, and metabolomic data. On the other hand, using genetic engineering tools such as gene knockouts, gene overexpression, or RNA interference, specific genes or metabolic pathways of both the fungus and the plant could be manipulated to study their roles in the symbiosis. Moreover, the development of fungal mutants could help to elucidate the functions of the specific genes involved in colonization, nutrient exchange, and the synthesis of bioactive compounds.

Therefore, to obtain the maximum profit from the use of EFs to face abiotic stress factors in crops, multidisciplinary work is necessary that addresses the problem from different perspectives to achieve short-term benefits with achievable and measurable goals.

## 8. Conclusions

Further research is necessary to explore the diversity of endophytes present in various ecosystems, focusing on those that have demonstrated the potential to improve the resilience of crops under environmental stresses such as drought, salinity, and nutrient deficiencies. It is necessary to consider the unique attributes of endophytes from extreme environments (e.g., deserts, saline soils) and their potential application in the development of stress-resistant crop varieties, as these endophytes are often well adapted to harsh conditions. The potential applications of endophytes not only for improving stress tolerance, but also for enhancing the overall plant fitness, growth, nutrient uptake, and yield, thereby providing comprehensive benefits to the crops, must be clearly delimited. Agricultural practices can be optimized to harness the full potential of EFs in improving crop resilience and productivity, especially to face environmental challenges and help hunger eradication. Harnessing the full potential of EFs in agricultural systems requires a multidisciplinary approach that integrates molecular biology, plant physiology, and ecological studies. By better understanding these symbiotic interactions, we can unlock new opportunities for sustainable agriculture in the face of climate-change-induced events.

## Figures and Tables

**Figure 1 microorganisms-12-01357-f001:**
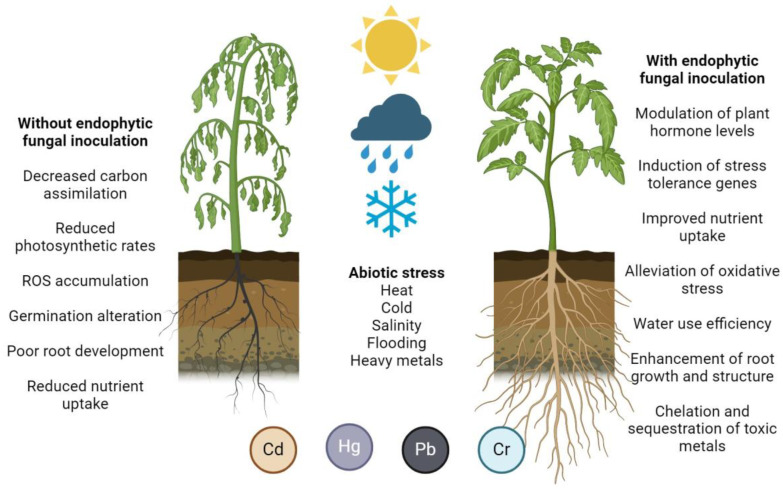
Plant benefits related to endophytic fungal association.

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
