# Peer review of "Endophytic Fungi for Crops Adaptation to Abiotic Stresses"

_microorganisms, 2024, doi:10.3390/microorganisms12071357_

Round 1

Reviewer 1 Report

Comments and Suggestions for Authors

Review: Endophytic fungi for crops adaptation to abiotic stresses

Overall, this is a relatively short attempt to summarize information on fungal endophytes and their effects on ameliorating abiotic stress. The current literature on the topic is not well referenced, and some important papers are not referred to at all e.g. Sodhi & Saxena 2023 Environmental and Experimental Botany 2023; Woo et al Nature Reviews Microbiology 2023

There is confusion right at the start of this paper: The abstract and the first sentence of the main part of the paper refer to EF as the abbreviation for endophytic fungi but then switch to EM as the abbreviation. Is EM different from EF?

L35 microscopic not microscopical

L68 locate not localize

L158 amelioration?

Figure 1: Check for spelling errors e.g. pathogen not phatogen; ‘Decreased’

L163 electrical conductivity: is ds/m the correct units?

L188 ..application of beneficial organisms like fungal endophytes… Give a reference for this. Unsure how EM could be applied successfully?

L190-207 Quite a major paragraph yet only two references given.

L208 ‘increase’

L212 -The reference quoted is very much focused on biotic stress

L282 Within the problems caused by toxic metals: fungal endophytes promoted as an alternative to reduce oxidative stress- based on a single reference. “impact on crop production is a red flag for hunger and malnutrition eradication” This does not make a lot of sense.

L289-290  Sentence needs to be clarified. What is meant by …metals flocculating molecules…

Comments on the Quality of English Language

The standard of the English is generally good. Some aspects, however, could certainly be improved. Most of these have been commented on in the Comments and Suggestions section.

Author Response

Overall, this is a relatively short attempt to summarize information on fungal endophytes and their effects on ameliorating abiotic stress. The current literature on the topic is not well referenced, and some important papers are not referred to at all e.g. Sodhi & Saxena 2023 Environmental and Experimental Botany 2023; Woo et al Nature Reviews Microbiology 2023

R= More relevant references were included in the manuscript.

There is confusion right at the start of this paper: The abstract and the first sentence of the main part of the paper refer to EF as the abbreviation for endophytic fungi but then switch to EM as the abbreviation. Is EM different from EF?

R=Addressed

The correct term is EF and it was changed.

L35 microscopic not microscopical

R=Addressed

L68 locate not localize

R=Addressed

L158 amelioration?

R=Addressed

Figure 1: Check for spelling errors e.g. pathogen not phatogen; ‘Decreased’

R=Addressed

L163 electrical conductivity: is ds/m the correct units?

R=Addressed

L188 ..application of beneficial organisms like fungal endophytes… Give a reference for this. Unsure how EM could be applied successfully?

R=Addressed

L190-207 Quite a major paragraph yet only two references given.

R=Addressed

L208 ‘increase’

R=Addressed

L212 -The reference quoted is very much focused on biotic stress

R=Addressed

L282 Within the problems caused by toxic metals: fungal endophytes promoted as an alternative to reduce oxidative stress- based on a single reference. “impact on crop production is a red flag for hunger and malnutrition eradication” This does not make a lot of sense.

R=Addressed

L289-290  Sentence needs to be clarified. What is meant by …metals flocculating molecules…

R=Addressed

Reviewer 2 Report

Comments and Suggestions for Authors

Dear authors,

 I have read your manuscript "Endophytic fungi for crops adaptation to abiotic stresses" carefully.  The field of research you described was interesting to me as a reader. But it seems to me inappropriate to publish the article as it is now. If you spend a little more time rethinking and rewriting the article, and make a submission again, article will be more useful for readers and become more cited. I will try to explain my point of view. Endophytic fungi are quite a popular object of research. Dozens of review articles and several meta-analyses can be found online, including those on the effect of endophytic fungi on plant stress resistance. Writing another review requires justification. This may be the publication of new research that changes the way we look at the problem, the emergence of new research methods, the establishment of interdisciplinary connections, or something else. Perhaps your own research will help you find a new point of view and make the review more original. Also the list of references contains too many review articles. Try to use more research articles. This is especially important in describing the gap in research.

I wish you successful and productive work.

Author Response

 I have read your manuscript "Endophytic fungi for crops adaptation to abiotic stresses" carefully.  The field of research you described was interesting to me as a reader. But it seems to me inappropriate to publish the article as it is now. If you spend a little more time rethinking and rewriting the article, and make a submission again, article will be more useful for readers and become more cited. I will try to explain my point of view. Endophytic fungi are quite a popular object of research. Dozens of review articles and several meta-analyses can be found online, including those on the effect of endophytic fungi on plant stress resistance. Writing another review requires justification. This may be the publication of new research that changes the way we look at the problem, the emergence of new research methods, the establishment of interdisciplinary connections, or something else. Perhaps your own research will help you find a new point of view and make the review more original. Also the list of references contains too many review articles. Try to use more research articles. This is especially important in describing the gap in research.

R=The article was rewritten and reorganized. Two sections were added to expand the topics and to establish perspectives for the study and applications of EF in crops production.

 I wish you successful and productive work.

R=Thank you, best regards.

Reviewer 3 Report

Comments and Suggestions for Authors

The study of the mechanisms of interaction between endophytic fungi and plants from the point of view of plant protection against environmental stresses is undoubtedly a very relevant area of modern biological science.

However, the presented review paper "Endophytic fungi for crops adaptation to abiotic stresses" is written superficially, the structure and content should be reconsidered.

In the Introduction part, I recommend focusing on the problem of abiotic stresses first and then on the application of beneficial endophytic fungi as an eco-friendly approach to improving crop adaptation. It should be clearly defined which endophytic fungi can be useful for plants.

After examples of the beneficial effects of Endophytic fungi on plants under abiotic stresses, then their mechanisms should be described (i.e., plant colonization, production of a wide range of biologically active compounds, induction of the host’s systemic resistance, etc.).

In the current version of the paper, no attention is paid to the effect of endophytic fungi on the microbial community of soils. This aspect also serves to consider and describe the known information.

Moreover, the paper does not allow us to understand exactly what gaps there are, what exactly should be paid attention to in future research and why, and what methods should be used to uncover the mechanisms of interaction between beneficial endophytic fungi and plants under abiotic stress conditions.

Other comments:

Lines 38, 41, 43, 46, 49, 51,…… It must be used “EF” instead of “EM”. Check it throughout the text.

Figure 1. It should be reconsidered and supplemented with more detailed information. For example, why only the “induction of salt tolerance genes”? Are tolerance genes to other abiotic stresses (drought, high/low temperatures, heavy metals, etc.) not induced? The review is devoted to abiotic stresses (according to the title of the article), but why are biotic stresses also included? The title should reflect the content of the paper.

Table 1. I recommend adding another column to indicate the specific stress for each example. Additionally, as many examples as possible should be given.

Subsection 5.3. “Crops protection against oxidative stress by endophytic fungi”. All abiotic stresses cause oxidative stress (damages) in plants, so the title of this subsection must be changed, for example, on ”Crops protection against toxic metals by endophytic fungi“, since this subsection is focused on Toxic Metals.

Also, another subsection about the influence of endophytic fungi on crop drought tolerance should be added. Because drought is one of the major abiotic stresses worldwide.

Comments on the Quality of English Language

Moderate editing of English language required.

Author Response

The study of the mechanisms of interaction between endophytic fungi and plants from the point of view of plant protection against environmental stresses is undoubtedly a very relevant area of modern biological science.

However, the presented review paper "Endophytic fungi for crops adaptation to abiotic stresses" is written superficially, the structure and content should be reconsidered.

In the Introduction part, I recommend focusing on the problem of abiotic stresses first and then on the application of beneficial endophytic fungi as an eco-friendly approach to improving crop adaptation. It should be clearly defined which endophytic fungi can be useful for plants.

R=The article was rewritten and reorganized to address this recommendation.

After examples of the beneficial effects of Endophytic fungi on plants under abiotic stresses, then their mechanisms should be described (i.e., plant colonization, production of a wide range of biologically active compounds, induction of the host’s systemic resistance, etc.).

R=More examples were added in the main text and in Table 1.

In the current version of the paper, no attention is paid to the effect of endophytic fungi on the microbial community of soils. This aspect also serves to consider and describe the known information.

R=One section was added with this purpose.

Moreover, the paper does not allow us to understand exactly what gaps there are, what exactly should be paid attention to in future research and why, and what methods should be used to uncover the mechanisms of interaction between beneficial endophytic fungi and plants under abiotic stress conditions.

R=The article was rewritten and reorganized to address this recommendation.

Other comments:

Lines 38, 41, 43, 46, 49, 51,…… It must be used “EF” instead of “EM”. Check it throughout the text.

R=Addressed

Figure 1. It should be reconsidered and supplemented with more detailed information. For example, why only the “induction of salt tolerance genes”? Are tolerance genes to other abiotic stresses (drought, high/low temperatures, heavy metals, etc.) not induced? The review is devoted to abiotic stresses (according to the title of the article), but why are biotic stresses also included? The title should reflect the content of the paper.

R=Figure 1 was reorganized and recreated.

Table 1. I recommend adding another column to indicate the specific stress for each example. Additionally, as many examples as possible should be given.

R=Addressed

Subsection 5.3. “Crops protection against oxidative stress by endophytic fungi”. All abiotic stresses cause oxidative stress (damages) in plants, so the title of this subsection must be changed, for example, on ”Crops protection against toxic metals by endophytic fungi“, since this subsection is focused on Toxic Metals.

R=Addressed 

Also, another subsection about the influence of endophytic fungi on crop drought tolerance should be added. Because drought is one of the major abiotic stresses worldwide.

R=Addressed.

Reviewer 4 Report

Comments and Suggestions for Authors

The purpose of this review is to provide a comprehensive overview of the current understanding of the role of endophytic fungi in enhancing crop adaptation to abiotic stress. Although the selected references are relevant to the topic, including many published recently (between 2020-2023), there is a need to explain at the end of topic 1 how this survey of articles was carried out, period, selection criteria, database, etc. The selected topics were appropriate and offered a broad view of the role of endophytic fungi in crop adaptation to abiotic stress. The manuscript contains one table and one figure. I believe that an illustration of topic 2 (mechanisms of endophytic fungi colonization of plant tissues) would make the text even more interesting and facilitate the understanding of this topic. Overall, the manuscript is well-written and concise. I believe that it is suitable for publication in your journal after minor revision.

Author Response

The purpose of this review is to provide a comprehensive overview of the current understanding of the role of endophytic fungi in enhancing crop adaptation to abiotic stress. Although the selected references are relevant to the topic, including many published recently (between 2020-2023), there is a need to explain at the end of topic 1 how this survey of articles was carried out, period, selection criteria, database, etc. The selected topics were appropriate and offered a broad view of the role of endophytic fungi in crop adaptation to abiotic stress. The manuscript contains one table and one figure. I believe that an illustration of topic 2 (mechanisms of endophytic fungi colonization of plant tissues) would make the text even more interesting and facilitate the understanding of this topic. Overall, the manuscript is well-written and concise. I believe that it is suitable for publication in your journal after minor revision.

R=The main of the commentaries were addressed. The article was rewritten and reorganized. Figure 1 was recreated. Table 1 was reorganized and expanded.

Round 2

Reviewer 2 Report

Comments and Suggestions for Authors

Dear authors, the manuscript has been significantly improved.

Author Response

Thank you, best regards.

Reviewer 3 Report

Comments and Suggestions for Authors

The authors have significantly revised the article and improved the presentation, but there are still points that require attention:

Introduction: paragraphs 2 and 3 repeat the information, they should be combined and shortened.

Lines 206-207. The list of beneficial EF used in agriculture must be reconsidered. Fusarium species commonly known as a dangerous causative agents of plant diseases. Be careful!

Lines 386, 404, 406, 408, 410, 412, 415...  “Endophytic fungi” use abbreviation EF as throughout the text.

Author Response

The authors have significantly revised the article and improved the presentation, but there are still points that require attention:

R=Thank you, best regards.

Introduction: paragraphs 2 and 3 repeat the information, they should be combined and shortened.

R=Addressed

Lines 206-207. The list of beneficial EF used in agriculture must be reconsidered. Fusarium species commonly known as a dangerous causative agents of plant diseases. Be careful!

R=A note was added about it 

Lines 386, 404, 406, 408, 410, 412, 415...  “Endophytic fungi” use abbreviation EF as throughout the text.

R=Addressed